# Knowledge Distillation in Object Detection: A Survey from CNN to Transformer

**DOI:** 10.3390/s26010292

**Published:** 2026-01-02

**Authors:** Tahira Shehzadi, Rabya Noor, Ifza Ifza, Marcus Liwicki, Didier Stricker, Muhammad Zeshan Afzal

**Affiliations:** 1Department of Computer Science, RPTU Kaiserslautern-Landau, 67663 Kaiserslautern, Germanydidier.stricker@dfki.de (D.S.); 2Mindgarage Lab, RPTU Kaiserslautern-Landau, 67663 Kaiserslautern, Germany; 3German Research Institute for Artificial Intelligence (DFKI), 67663 Kaiserslautern, Germany; 4Department of Computer Science, Electrical and Space Engineering Luleå University of Technology, 971 87 Luleå, Sweden

**Keywords:** transformer, knowledge distillation, DETR, computer vision, deep neural networks

## Abstract

Deep learning models, especially for object detection have gained immense popularity in computer vision. These models have demonstrated remarkable accuracy and performance, driving advancements across various applications. However, the high computational complexity and large storage requirements of state-of-the-art object detection models pose significant challenges for deployment on resource-constrained devices like mobile phones and embedded systems. Knowledge Distillation (KD) has emerged as a prominent solution to these challenges, effectively compressing large, complex teacher models into smaller, efficient student models. This technique maintains good accuracy while significantly reducing model size and computational demands, making object detection models more practical for real-world applications. This survey provides a comprehensive review of KD-based object detection models developed in recent years. It offers an in-depth analysis of existing techniques, highlighting their novelty and limitations, and explores future research directions. The survey covers the different distillation algorithms used in object detection. It also examines extended applications of knowledge distillation in object detection, such as improvements for lightweight models, addressing catastrophic forgetting in incremental learning, and enhancing small object detection. Furthermore, the survey also delves into the application of knowledge distillation in other domains such as image classification, semantic segmentation, 3D reconstruction, and document analysis.

## 1. Introduction

Knowledge Distillation (KD) [1] is a pivotal technique in deep learning for transferring knowledge from large, high-capacity teacher models to smaller, more efficient student models. Instead of relying solely on hard labels, the student leverages the teacher’s outputs, including soft probability distributions and intermediate representations, as guidance. The objective is to produce a model that is computationally lightweight while retaining strong performance, making KD especially valuable for deployment on resource-constrained devices.

Knowledge Distillation (KD) has been widely adopted across various domains to enhance efficiency and performance of AI models. In natural language processing, it supports few-shot learning for intent classification and compresses large language models for deployment efficiency [2]. Meanwhile, in speech recognition, it compresses models for mobile and edge devices without significant accuracy loss [3]. The medical field also benefits from knowledge distillation in medical imaging for tasks like tumor detection and classification. Moreover, other healthcare applications include predictive models for patient outcomes and disease diagnosis in resource-constrained environments [4]. KD is also utilized by autonomous vehicles for domain adaptive semantic segmentation, improving their safety and performance [5]. In agriculture, it aids in crop disease detection and yield prediction [6]. Additionally, smart city initiatives employ knowledge distillation for traffic management, surveillance, and resource allocation, making AI solutions more scalable and accessible [7]. These diverse applications underscore the versatility and transformative impact of knowledge distillation across various fields.

Similarly, knowledge distillation techniques have been instrumental in advancing the field of computer vision. In image classification, KD helps create lightweight models that maintain high accuracy, facilitating deployment on mobile and embedded devices [8,9,10,11,12,13,14]. For semantic segmentation, KD enables the development of efficient models [15,16,17,18,19,20,21,22,23,24,25,26,27,28,29,30] capable of precise pixel-wise predictions, essential for tasks such as autonomous driving [5]. In the realm of 3D reconstruction [31,32,33], KD aids in reducing the computational complexity of models, allowing for real-time processing and accurate reconstruction of 3D scenes from 2D images [34,35,36,37]. Document analysis also benefits from KD, where it helps in developing compact models that can efficiently extract and interpret information from scanned documents, enhancing OCR systems and automated document processing [38,39,40,41]. However, applying KD in object detection is more challenging compared to image classification due to the need to accurately localize and classify multiple objects within an image. This survey focuses on the applications of KD in object detection, exploring how this technique has been leveraged to improve the efficiency and accuracy of object detection models in various practical scenarios.

Object detection is a core computer vision task that involves identifying and localizing objects within an image by predicting both their categories and bounding boxes. The field has advanced significantly, evolving from classical methods to modern deep learning-based approaches [42,43,44,45,46,47,48,49,50,51,52,53,54,55,56,57,58,59,60,61]. Early approaches relied on edge detection and handcrafted features [62,63,64], later improved through machine learning techniques such as HOG descriptors and SVMs [65,66]. The breakthrough came with deep neural networks, beginning with two-stage detectors like Fast R-CNN [67] and Faster R-CNN [43], which combined convolutional networks with region proposal methods for improved accuracy and speed. Subsequently, one-stage detectors such as YOLO [46] and SSD [48] prioritized real-time performance, while enhancements like Feature Pyramid Networks (FPNs) [45] improved multi-scale detection. Extensions such as Mask R-CNN [49] added instance segmentation capabilities, and EfficientDet [44] further optimized accuracy–efficiency trade-offs through compound scaling. More recently, transformer-based detectors, including Vision Transformers (ViTs) [50] and Detection Transformers (DETR) [51], introduced self-attention to model global object relationships, pushing the boundaries of detection performance. Emerging directions such as self-supervised and few-shot learning aim to reduce reliance on large labeled datasets. In summary, modern object detectors can be broadly categorized into three groups: two-stage, one-stage, and transformer-based, as shown in Figure 1.

The application of knowledge distillation is crucial in the domain of object detection to address several key limitations. High computational complexity makes deployment on resource-constrained devices difficult [68]. Moreover, achieving real-time processing is challenging due to significant computational demands [69]. Furthermore, generalization to diverse environments is problematic, leading to decreased accuracy [69].

Additionally, training robust detectors requires large amounts of labeled data, which is costly and hard to obtain [68]. Knowledge distillation addresses these issues by transferring knowledge from larger, high-performing models to smaller, efficient models, reducing computational requirements, enabling real-time processing, enhancing generalization, and improving performance even with less labeled data [70]. To ensure a systematic and reproducible review, we adopted a structured process for collecting and organizing the literature, as in Figure 1. We included studies that introduced new knowledge distillation (KD) methods or applied KD to object detection with evaluations on standard benchmarks, while excluding works that lacked methodological novelty or experimental results. The selected studies were categorized by architecture type as CNN-based and Transformer-based. This organization provides a consistent structure, facilitating replication and future extensions of our review. Building on this, we provide a comprehensive overview of the latest KD architectures for object detection, highlighting the diverse methodologies and innovations designed to improve model efficiency and performance. We examine different distillation algorithms, including response-based [1,71,72,73,74], feature-based [75,76,77,78,79,80,81,82,83,84,85,86,87,88,89], relation-based [90,91,92,93,94,95,96,97,98,99,100,101,102], task-specific [103,104], and self-distillation [90,105]. We further discuss KD applications beyond object detection, including image classification [106,107,108,109,110,111,112], semantic segmentation, 3D reconstruction, and document analysis. We also discuss the limitations and challenges that currently hinder the widespread adoption of KD, and conclude by identifying future research directions for knowledge distillation in object detection.

## 2. Applications of Knowledge Distillation in Object Detection

### 2.1. CNN Based

CNN-based KD methods have demonstrated significant improvements in efficiency and accuracy for object detection. Feature-based approaches (e.g., SKD [113], PKD [114]) focus on aligning structural or relational representations, while response-based methods (e.g., DKD [115], MLLD [116]) emphasize logit alignment. Some strategies, such as ERD [117] and ICD [118], target incremental learning and instance-specific distillation, whereas others (e.g., KD-Zero [119], Auto-KD [14]) automate the design of KD strategies.

#### 2.1.1. HEAD

Wang et al. [120] address the challenge of knowledge distillation between heterogeneous object detectors with the HEtero-Assists Distillation (HEAD) framework. HEAD introduces an assistant head to the student detector that is architecturally identical to the teacher’s head, transforming the distillation process into a homogeneous one for more effective knowledge transfer [71,75,121,122,123,124,125]. The framework consists of two key components: Assistant-based Knowledge Distillation (AKD) and Cross-architecture Knowledge Distillation (CKD). In AKD, the assistant head learns from the teacher head, while CKD provides direct supervision from the teacher to the student. As illustrated in Figure 2, a Faster R-CNN [43] teacher transfers knowledge to a RetinaNet [47] student. The assistant head processes features from the student backbone, overcoming the semantic gap between heterogeneous detectors. By leveraging the strengths of both architectures, HEAD significantly improves the performance of the student model.

#### 2.1.2. SSD-Det

Wu et al. [126] proposes SSD-Det: a Spatial Self-Distillation based Object Detector. Due to several reasons such as human error [127], difficulty in annotating accurate bounding boxes without background knowledge of the domain [128,129] among others [130], it is vital to address the issue of object detection with inaccurate bounding boxes [129,131,132,133,134]. This is tackled through the SSD-Det. The architecture incorporates two key modules: Spatial Position Self-Distillation and Spatial Identity Self-Distillation. In SSD-Det, Knowledge Distillation (KD) is implemented through a self-distillation process. Spatial Position Self-Distillation refines initial proposals using spatial information to create proposal bags. The model utilizes its own spatial cues, derived from intermediate layers, to guide the refinement of these proposals, essentially learning from itself. Spatial Identity Self-Distillation predicts the IoU between objects and proposals to enhance proposal selection. This involves the model using its own output to teach itself, improving its ability to select the best proposals based on the predicted IoU. This interactive structure of combining spatial and category information allows for better handling of inaccurate annotations. Figure 3 illustrates the process, showing how both modules refine the bounding boxes iteratively, leading to more accurate object detection.

#### 2.1.3. HierKD

Ma et al. [135] proposes a method to enhance open-vocabulary [136] one-stage object detection through Hierarchical Visual-Language Knowledge Distillation (HierKD). Traditional two-stage detectors use instance-level visual-to-visual knowledge distillation [136,137,138,139,140,141], but this approach is less effective for one-stage detectors due to the lack of class-agnostic proposals. HierKD addresses this by combining instance-level knowledge distillation (IKD) with global-level knowledge distillation (GKD), as shown in Figure 4. IKD transfers knowledge from positive sample points on feature maps, while GKD aligns multi-layer feature maps with captions from a pre-trained visual-language model [142] using cross-attention mechanisms. This hierarchical approach ensures that semantic knowledge from both seen and unseen categories is effectively transferred, significantly improving detection performance for novel categories.

#### 2.1.4. ERD

Feng et al. [117] propose Elastic Response Distillation (ERD) to address catastrophic forgetting in incremental object detection, where models often lose previously learned knowledge while adapting to new object classes [106,143]. ERD tackles this by elastically learning responses from both the classification and regression heads of the teacher model. Using Elastic Response Selection (ERS), the method evaluates and selects the most informative responses, ensuring the student model retains crucial knowledge while effectively learning new information. Category and localization knowledge are transferred through selective distillation [144], as illustrated in Figure 5. This approach significantly reduces the performance gap between incremental and full training by focusing on the most valuable responses.

#### 2.1.5. MLLD

Jin et al. [116] propose a framework, Multi-Level Logit Distillation (MLLD), that enhances the distillation process by aligning teacher and student predictions at multiple levels thereby addressing the performance gap and privacy issues in traditional logit distillation [145,146,147,148,149]. The architecture in Figure 6, involves three main components: prediction augmentation, multi-level alignment, and logit alignment.

The framework begins with the teacher and student models processing the same input image. The prediction augmentation component generates enhanced predictions from both models. These predictions are then aligned at three levels: instance-level, batch-level, and class-level. This multi-level alignment ensures that the student model not only mimics the teacher model’s outputs closely but also learns the intricate patterns and distributions present in the teacher’s predictions.

#### 2.1.6. FAM

Pham et al. [150] introduce a method called Frequency Attention Module (FAM) for Knowledge Distillation, which leverages the frequency domain to enhance the distillation process. Previously, attention-based knowledge distillation methods [77,151,152] primarily operated in the spatial domain, focusing on local regions of the input feature maps. This spatial attention can miss out on capturing broader contextual information necessary for effective knowledge transfer. FAM, as depicted in Figure 7, uses a learnable global filter that adjusts the frequency components of the student’s feature maps based on the teacher’s features. This adjustment helps the student model to better mimic the teacher model’s patterns. The process involves transforming the feature maps into the frequency domain using Fast Fourier Transform (FFT) [153], applying the learnable filter, and then transforming them back to the spatial domain using inverse FFT (IFFT). This method ensures that both local and global information is effectively captured and transferred. The proposed method significantly improves the performance of student models on both image classification and object detection tasks by effectively capturing and transferring the hierarchical patterns present in the teacher model’s features.

#### 2.1.7. SKD

Rijk et al. [113] proposes a novel approach called Structural Knowledge Distillation (SKD) for object detection, as shown in Figure 8. Feature-based distillation methods typically rely on minimizing the lp-norm between teacher and student feature activations, which often ignores the spatial relationships and structural information inherent in the features [69,118,121,122,123,125].

SKD addresses this by incorporating structural similarity (SSIM) into the distillation process, which captures luminance, contrast, and structural information between teacher and student features. The architecture, depicted in Figure 8, involves decomposing feature spaces into local patches and computing SSIM components for these patches. This method ensures that spatial dependencies and feature correlations are preserved during distillation hence enhancing the student’s ability to replicate the teacher’s performance more effectively than traditional methods.

#### 2.1.8. ICD

Kang et al. [118] presents Instance-Conditional Knowledge Distillation (ICD), which enhances knowledge transfer by considering the unique contributions of different instances. Traditional methods for knowledge distillation in object detection often use region-based sampling [71,121,122,123,154] but struggle to balance the varying contributions of different features, making distillation less effective. The ICD framework, illustrated in Figure 9, introduces a conditional decoding module that retrieves knowledge based on instance conditions, such as category and location. This module computes instance-conditional knowledge by measuring the correlation between instance queries and the teacher’s representations using a transformer-based attention mechanism. An auxiliary task is used to optimize the decoding process, ensuring that the student model learns both identification and localization effectively. This approach improves the distillation process by explicitly focusing on the most informative features for each instance, leading to better detection performance.

#### 2.1.9. PCD

Huang et al. [155] proposes Pixel-Wise Contrastive Distillation (PCD), which uses distillation in a self-supervised manner to enhance prediction tasks in dense detectors [156] at the pixel level taking inspiration from contrastive learning [157]. PCD includes a SpatialAdaptor, which preserves the output feature’s distribution despite also processing the feature maps through modification of the teacher’s projection head. In this framework, the effective receptive field of the student is improved using self attention with multiple heads [158]. The process involves matching corresponding pixels between the feature maps of both the student as well as the teacher and contrasting them with negative samples from a memory queue [159]. This method, as depicted in Figure 10, significantly boosts the performance of small models on dense prediction tasks by ensuring a more detailed pixel-level knowledge transfer from teacher to student.

#### 2.1.10. DKD

Zhao et al. [115] propose Decoupled Knowledge Distillation (DKD) to enhance logit-based knowledge distillation [145] by separating target class knowledge distillation (TCKD) and non-target class knowledge distillation (NCKD).Traditional KD methods couple these components, which suppresses NCKD’s effectiveness and limits flexibility. DKD reformulates the classical KD loss into two distinct parts: TCKD and NCKD. DKD reformulates the KD loss to decouple TCKD and NCKD, allowing independent adjustment of their contributions. This decoupling improves training efficiency and performance, as shown in Figure 11, and demonstrates superior results on datasets for image classification and object detection.

#### 2.1.11. KD-Zero

Li et al. [119] introduces KD-Zero, a framework designed to automate the search for optimal knowledge distillation strategies using evolutionary algorithms. This framework, as shown in Figure 12, addresses challenges such as the capacity gap [147,160,161] between teacher and student models, cross-architecture compatibility [1,147,160,161], and the inefficiency of manual tuning. KD-Zero decomposes distillation into knowledge transformations, distance functions, and loss weights, and employs evolutionary search to discover the optimal combination of these components.

The search process is accelerated with loss-rejection protocols and search space shrinkage, ensuring efficiency. The architecture, depicted in Figure 12, shows how KD-Zero initializes, evaluates, and evolves candidate distillers to optimize performance across various tasks, significantly improving the distillation process without the need for expert intervention.

#### 2.1.12. FGD

Yang et al. [124] propose a method to improve object detection by addressing the imbalance between foreground and background features, a key challenge in this domain [47]. To tackle this, they introduce Focal and Global Distillation (FGD), designed to overcome limitations of previous approaches [71,122,123,154]. FGD has two main components: focal distillation and global distillation [162]. Focal distillation separates foreground and background regions, guiding the student model to focus on the teacher’s important pixels and channels. Global distillation captures relationships between all pixels, providing the student with broader contextual information that complements focal distillation. Figure 13 illustrates this mechanism.

#### 2.1.13. UniKD

Lao et al. [163] propose Universal Knowledge Distillation (UniKD), which introduces Adaptive Knowledge Extractor (AKE) modules to bridge the semantic gap between heterogeneous teacher–student pairs. Unlike traditional feature-based distillation methods that struggle with cross-architecture compatibility [120,164,165], UniKD uses AKEs with deformable cross-attention to distill knowledge effectively. The AKE modules first absorb detection-relevant knowledge from the teacher model and then transfer it to the student model. This approach, depicted in Figure 14, ensures the student model benefits from the high-level knowledge of the teacher, regardless of architectural differences. UniKD’s query-based paradigm not only reduces storage costs but also significantly improves detection performance for both homogeneous and heterogeneous pairs.

#### 2.1.14. DiffKD

Huang et al. [166] proposes DiffKD, which leverages diffusion models [167,168,169] to reduce noise in student features and improve knowledge transfer from teacher models. DiffKD addresses the challenges faced by previous methods due to the capacity gap between teacher and student models, resulting in noisy student features. It does this by treating student features as noisy versions of teacher features and using a diffusion model to denoise them. Figure 15, includes a light-weight diffusion model with bottleneck blocks and a linear auto encoder to reduce computation costs. The adaptive noise matching module further enhances denoising performance by adjusting the noise level in the student features. Thus this approach ensures that only valuable information is distilled and can be used in image classification, object detection, and semantic segmentation as well.

#### 2.1.15. Auto-KD

Li et al. [14] propose an automated search framework for optimal knowledge distillation design as opposed to previous distillation methods [77,95], which often rely on handcrafted architectures and extensive tuning, making it complex [170], and are specific to teacher–student pairs. Auto-KD addresses these issues by decomposing distillers into basic operations and using a Monte Carlo Tree Search (MCTS) [171] to explore the search space efficiently. The framework integrates advanced operations like transformations, distance functions, and hyperparameters into a unified tree-structured search space. The architecture in Figure 16, includes offline processing, sparse training, and proxy settings to accelerate the search process. This structure ensures that the optimal distillation strategies are identified with minimal computational overhead. Auto-KD’s approach significantly improves performance across various vision tasks, including image classification, object detection, and semantic segmentation.

#### 2.1.16. LD

Zheng et al. [172] propose a method, Localization Distillation (LD) to enhance object detection by focusing on localization knowledge distillation. Unlike previous methods that primarily mimic classification features [118,123,125], this approach uses a probability distribution for bounding boxes [173,174] to transfer rich localization information from teacher to student models. They also introduce valuable localization regions (VLRs) to selectively distill both semantic and localization knowledge, improving overall performance. Figure 17 illustrates the architecture, where region weighting helps determine the main distillation regions and VLRs, and the localization head converts bounding boxes to probability distributions, with the loss calculated to align the teacher and student models effectively.

#### 2.1.17. DIST

Huang et al. [160] introduces Distillation from a Stronger Teacher, which focuses on improving the performance of student models by leveraging stronger teachers. KD methods often struggle with significant discrepancies between teacher and student models, particularly when the teacher is much stronger [145,147,175]. DIST addresses this by using a correlation-based loss instead of the traditional KL divergence [1] to match the predictions of the teacher and student models. This method preserves the relational information between classes and within each class, allowing the student to benefit from the teacher’s stronger performance without needing to match predictions exactly. The architecture, shown in Figure 18, illustrates how DIST preserves both inter-class and intra-class relationships, ensuring that the student model effectively captures the essential knowledge from the teacher.

This approach results in improved performance across image classification, object detection, and semantic segmentation tasks.

#### 2.1.18. CTCP

Yang et al. [176] propose a method to address protocol inconsistencies (CTCP) in knowledge distillation for dense object detectors. This approach includes two novel distillation losses, namely IoU-based Localization Distillation Loss, and Binary Classification Distillation Loss. Binary Classification Distillation reformulates classification logits [172] into multiple binary-classification maps, addressing the imbalance between foreground and background samples [47] by using a sigmoid protocol. IoU-based Localization Distillation directly computes the IoU between teacher and student bounding boxes, enabling effective localization knowledge transfer without specific head structures. This combined approach, as illustrated in Figure 19, enhances both classification and localization performance by aligning student outputs more closely with teacher predictions.

#### 2.1.19. SED

Guo et al. [177] addresses challenges in semi-supervised object detection, such as large variance in object sizes and class imbalance, by introducing Scale-Equivalent Distillation (SED). This method imposes a consistency regularization to handle scale variance, which was not considered by previous Semi-Supervised Object Detection methods [178,179,180,181], alleviates noise from false negatives and localization errors, and uses a re-weighting strategy to focus on potential foreground regions. The architecture is shown in Figure 20.

It includes a teacher–student framework where the teacher model, updated via exponential moving average, generates soft pseudo-labels for the student. The student processes strongly augmented images, ensuring consistency across different scales and reducing the impact of class imbalance. This approach significantly improves detection performance, demonstrating robustness to size variance and class imbalance.

#### 2.1.20. ScaleKD

In Scale-aware Knowledge Distillation (ScaleKD) [182], the authors introduce a framework designed to improve the performance of small object detectors [43,47,52,183,184] through scale-aware knowledge distillation. The architecture, as shown in Figure 21, extracts multi-scale features from high-resolution images using the teacher network and aligns them with features from the student network processing standard-resolution images. A scale-aware distillation loss ensures effective learning across different object sizes. The Cross-Scale Assistant module aids in aligning features between teacher and student at various stages, enhancing detection accuracy. This method significantly boosts performance for small object detectors by addressing scale variance challenges.

#### 2.1.21. PKD

Cao et al. [114] propose a novel distillation framework called Pearson Correlation Coefficient-based Knowledge Distillation (PKD) to enhance the knowledge transfer process in object detection, particularly addressing the challenges posed by heterogeneous teacher–student detector pairs [124,125]. Traditional distillation methods often struggle with the magnitude differences in feature maps, leading to sub-optimal results [69,71,122,124]. PKD mitigates this by using the Pearson Correlation Coefficient to focus on the relational information between features rather than their absolute values. The architecture, shown in Figure 22, normalizes the feature maps from both the teacher and student models to have zero mean and unit variance before applying the distillation loss.

This approach minimizes the MSE between the normalized features, effectively using the Pearson Correlation Coefficient to guide the distillation process. By doing so, PKD aligns the feature representations from the teacher and student, ensuring better knowledge transfer and improved detection performance.

### 2.2. Transformer Based

While CNN-based distillation methods [116,117,118,120,135] have laid a strong foundation for object detection, their representational capacity is limited by local receptive fields. Transformers, in contrast, employ self-attention mechanisms that capture global dependencies, making them better suited for transferring semantic and relational knowledge across objects. This shift from CNN-based to Transformer-based approaches reflects a broader trend in computer vision: moving from spatially localized representations toward globally contextualized knowledge. As a result, Transformer-based distillation methods often achieve better generalization, particularly in open-vocabulary detection and cross-domain adaptation.

#### 2.2.1. KDEP

He et al. [185] presents Knowledge Distillation as Efficient Pre-training (KDEP), a method that aims to streamline the pre-training [186] process by transferring feature extraction capabilities from pre-trained teacher models to student models. Normally pre-training demands large datasets [187,188,189,190] and computational resources. KDEP addresses this by using non-parametric alignment techniques like Singular Value Decomposition (SVD) [191] and Power Temperature Scaling (PTS) [192] to align feature dimensions between teacher and student models without adding extra parameters. The architecture, shown in Figure 23, effectively compresses and scales the features to maintain their integrity in the student model. This approach achieves faster convergence, higher data efficiency, and improved transfer-ability compared to conventional supervised pre-training.

#### 2.2.2. ZeroShot

Liu et al. [193] introduces an approach to apply distillation on features specifically for zero-shot object detection. This approach adapts the feature space of large models like CLIP [142] to the target detection domain. The adaptation is achieved through fine-tuning normalization layers, thereby bridging the domain gap. The method also introduces CLIP Proposals, which are image regions likely to contain novel categories, selected based on high objectness scores calculated using CLIP’s vision encoder. These regions facilitate efficient knowledge transfer without additional data. Figure 24 illustrates the architecture where the backbone network processes features, and the RoI head distills knowledge through a combination of the regressor and proposals to enhance detection performance.

#### 2.2.3. Forget

Kang et al. [194] addresses the issue of catastrophic forgetting in incremental learning [195]—specifically object detection as little work has been done in this domain [117,196]—by proposing a method that uses knowledge distillation both within class as well as between class. The framework, shown in Figure 25, incorporates two main components: Distance Matrix Distillation (DMD) and Interactive Feature Distillation (IFD). DMD preserves class-level distinctions by distilling class-wise semantic and feature distances from the teacher to the student. IFD maintains within-class consistency by ensuring instance-wise feature similarity between the teacher and student models. This dual approach allows the student model to retain knowledge of previously learned categories while integrating new ones, thus mitigating the problem of catastrophic forgetting.

#### 2.2.4. OADP

In the paper, Wang et al. [197] propose an Object-Aware Distillation Pyramid (OADP) framework that enhances open-vocabulary [136] object detection by focusing on object-aware feature distillation. The architecture in Figure 26 integrates a multi-level distillation process using a pyramid structure. The framework employs CLIP visual encoders to process high-resolution images, extracting object-aware features at different levels: block head, global head, and object head. These features are then distilled into the student model through specific distillation losses for each level ($L_{B}$, $L_{G}$, $L_{O}$), ensuring comprehensive knowledge transfer. The framework leverages a pyramid structure and integrates Region Proposal Network (RPN) and ROI Align mechanisms to handle object detection tasks, supported by object-level supervision.

#### 2.2.5. DETRDistill

DETRDistill [198] introduces a method to enhance transformer-based object detectors (DETRs) [199,200] through knowledge distillation, as shown in Figure 27. The framework includes three key components: Hungarian-matching logits distillation [1], target-aware feature distillation [201], and query-prior assignment distillation [198]. The Hungarian-matching aligns teacher and student predictions for effective knowledge transfer. Target-aware feature distillation uses object-centric features to create soft masks, refining the student’s focus on important regions. Query-prior assignment distillation leverages the teacher’s well-trained queries to stabilize the student’s learning process, enhancing convergence and performance. This architecture ensures comprehensive distillation and improved performance for DETR models.

#### 2.2.6. DK-DETR

Li et al. [202] introduces an approach to transfer the distilled visual-linguistic knowledge into the proposed DK-DETR from VLMs [142,203]. Previous methods for open-vocabulary [136] object detection [135,138,204] focused on transferring knowledge from VLMs to detectors using vanilla knowledge distillation, which often led to limited improvements and could degrade base category performance. In contrast, DK-DETR, as shown in Figure 28, introduces two novel distillation schemes: Semantic Knowledge Distillation (SKD) and Relational Knowledge Distillation (RKD). SKD aligns feature representations between the detector and VLM, treating feature alignment as a pseudo-classification problem rather than regression. RKD captures relationships between objects by modeling pair-wise object similarities. An auxiliary distillation branch with additional queries is introduced to prevent performance degradation of base categories during training. This combined approach effectively enhances the detection of novel categories while maintaining the performance on base categories.

Transformer-based KD approaches leverage global self-attention to transfer knowledge more effectively across objects and categories. Methods such as KDEP streamline pre-training, DETRDistill and DK-DETR enhance query-based detection, while Forget and OADP address incremental learning and open-vocabulary challenges. These approaches highlight the potential of Transformers to capture richer semantic and relational knowledge compared to CNNs. However, they typically require higher computational resources and careful optimization to balance performance with efficiency.

## 3. Applications of Knowledge Distillation

Knowledge distillation has found extensive applications across various domains, including 3D reconstruction, semantic segmentation, classification, document analysis and the medical domain [205,206,207,208].

### 3.1. Image Classification

Knowledge distillation has been extensively used in the domain of image classification where a smaller, more efficient student model learns to replicate the performance of a larger, more complex teacher model. An advancement in this field is the use of multi-teacher knowledge distillation, where multiple teacher models provide diverse insights, enhancing the student model’s learning experience. Additionally, multi-teacher distillation mitigates the error avalanche effect, where inaccuracies from a single teacher can cascade and amplify in the student model. Significant work has been done [8,9,10,11,12,13,14] that leverages this technique that leverages the strengths of multiple teachers to create a more effective and efficient student model. Moreover, capacity gap has also been addressed in image classification when applying knowledge distillation [23,115,160,209,210,211,212,213,214]. Capacity gap refers to the difference in the representational capacity between the teacher and the student models. The teacher model, being larger and more complex, can capture and represent intricate patterns and nuances in the data that the student is unable to capture, thus leading to a loss in accuracy.

### 3.2. Semantic Segmentation

Semantic segmentation involves classifying each pixel in an image into a predefined category or class. Recent progress in knowledge distillation (KD) has seen methods adapted specifically for this task. One of the techniques, structural knowledge distillation [15,16], transfers knowledge in segmentation tasks by using pair-wise similarities and holistic adversarial enhancements from a large teacher network to a compact student network. Another approach [17] optimizes the distillation process by focusing on global semantic interconnections across different images. Masked generative models [18] distill knowledge by guiding the student model in recovering missing features based on the teacher’s outputs. Channel-wise knowledge distillation [19] concentrates on channel-wise knowledge transfer through salient features within each channel between the teacher and student networks. ZeroSeg [20] explores the use of open-vocabulary models like CLIP for semantic segmentation by distilling knowledge from the vision encoder alone to improve segmentation tasks without extensive retraining. Boundary privileged knowledge distillation [21] focuses on refining the quality of object boundaries and leveraging shape constraints for inner regions. These are some of the most prominent approaches used to optimize knowledge transfer in semantic segmentation among others [22,23,24,25,26,27,28,29,30].

### 3.3. Three-Dimensional Reconstruction

Three-Dimensional reconstruction involves creating a three-dimensional model of an object or scene from multiple images or views. Knowledge distillation has significantly enhanced this domain by improving model efficiency and accuracy while reducing computational overhead. One recent approach, TempDistiller [34], captures long-term temporal memory and inter-frame relations to balance precision and speed in 3D object detection. Another technique, IFKD [35], proposed to enhance single-view 3D reconstructions by transferring knowledge from complex models to simpler ones. Moreover, another study [36] focuses on pillar and voxel-based detectors, achieving efficiency gains and accuracy through model compression and input resolution reduction. Lastly, PointDistiller [37] addresses point cloud sparsity and irregularity, improving the efficiency and compactness of 3D detection models. These advancements demonstrate the critical role of knowledge distillation in making 3D reconstruction models more practical and deployable across various applications.

### 3.4. Document Analysis

Document analysis [215,216,217,218,219] involves the extraction and processing of information from digital or scanned documents to understand their content, structure, and layout. Knowledge distillation is applied to create efficient, smaller models while retaining the performance of larger, more complex models. In DistilDoc [38], knowledge distillation techniques are applied to improve efficiency in document layout analysis and image classification, focusing on creating lean, high-performing models. GraphKD proposes a graph-based KD framework that constructs structured graphs to enhance document object detection while maintaining topological integrity. Another approach [40] focuses on simplifying the TinyBERT model to improve document retrieval efficiency, achieving significant speedups while maintaining performance. One more approach [41] explores the optimization of text–image machine translation models as different teacher models guide the optimization of various sub-modules within the end-to-end model, ensuring comprehensive knowledge transfer and improvements in translation accuracy. These applications of knowledge distillation in document analysis fall into categories such as document layout analysis, document object detection, form document understanding, multi-modal document understanding, and document retrieval.

## 4. Performance Results

Table 1 summarizes the performance of knowledge distillation (KD) applied to various object detection methods on the MS COCO [220] dataset. The table categorizes the methods into one-stage, two-stage, and end-to-end detectors, with performance measured using mAP, $AP_{50}$, and $AP_{75}$. Generally, two-stage detectors achieve higher performance than one-stage detectors when using the same KD strategy, as observed in methods such as ScaleKD, Auto-KD, and DiffKD. Among one-stage detectors, PKD achieves the highest performance, whereas ScaleKD outperforms all other methods among two-stage detectors.

Figure 29 provides a visual comparison of these methods. Panel (a) shows $AP_{50}$, and panel (b) shows mAP, with one-stage, two-stage, and end-to-end methods represented by circles, squares, and diamonds, respectively. The graphs confirm that two-stage detectors generally outperform one-stage detectors under the same KD strategy, while end-to-end methods demonstrate competitive performance comparable to two-stage detectors. These results highlight the effectiveness of KD across different detection paradigms.

Beyond raw AP metrics, the surveyed KD methods reveal distinct trade-offs. For instance, methods such as ScaleKD [182] and LD [172] excel at small-object detection and localization tasks, while ERD [117] and Forget are particularly effective in incremental learning scenarios. Automated search strategies such as KD-Zero [119], Auto-KD [14] reduce the burden of manual design but may introduce additional computational overhead. Feature-based approaches like SKD [113] and PKD [114] are lightweight and effective for cross-architecture distillation, whereas diffusion-based methods as DiffKD [166] provide strong accuracy gains at the cost of higher training complexity.

This comparative perspective emphasizes that the choice of KD strategy should be guided not only by accuracy but also by application requirements such as latency, memory, and robustness to domain shift.

## 5. Limitations

Knowledge distillation (KD) has shown considerable promise in various computer vision domains. In object detection, however, applying KD presents unique challenges. Unlike classification, object detection involves multiple tasks, classification, localization, and sometimes segmentation, making the distillation process more complex. Designing loss functions that effectively transfer both class and localization knowledge, while managing class imbalances and spatial dependencies [176], is particularly challenging.

Different categories of object detection methods introduce additional considerations. One-stage methods, such as HEAD [120] and FGD [124], are typically faster and more straightforward, directly predicting bounding boxes and class probabilities. Their simplicity allows easier integration of distillation strategies, but they often struggle with localization precision and may rely heavily on precise ground-truth boxes. Two-stage methods, including MLLD [116] and ERD [117], offer higher accuracy due to explicit region proposals and multi-level feature alignment. They are effective for preserving relational knowledge and supporting incremental learning, but their multi-stage pipelines increase computational complexity and hyperparameter sensitivity, which can limit scalability. Transformer-based methods, such as DK-DETR [202], DETRDistill [198], and DiffKD [166], excel at capturing global context and long-range dependencies, improving performance in open-vocabulary and cross-domain detection. However, these methods are often computationally expensive, sensitive to architectural mismatches, and may require large-scale pretraining or complex auxiliary modules.

Table Table 2 complements this discussion by summarizing the advantages and limitations of individual methods, helping researchers identify appropriate techniques while highlighting areas where future work can improve efficiency, robustness, and generalizability.In addition to these category-specific limitations, several challenges remain across all methods. The transfer of localization knowledge remains difficult, especially when the teacher and student differ significantly in capacity. Cross-architecture distillation continues to be a bottleneck, limiting the generalizability of many approaches. Computational efficiency is another concern, with methods like DiffKD [166] and HierKD [135] introducing significant overhead. Finally, multi-level and relational distillation strategies often involve intricate hyperparameter tuning, which can reduce practicality in large-scale or industrial settings.

## 6. Conclusions

In this survey paper, we have extensively reviewed and summarized recently proposed architectures that utilize knowledge distillation to enhance object detection. We have provided a detailed comparison of their performance and outlined the specific limitations associated with each approach. Our analysis offers a comprehensive understanding of the current state of research in this domain and highlights the significant advancements made possible through knowledge distillation techniques. The discussion within this paper underscores the significance of knowledge distillation in the field of object detection. It significantly enhances the field of object detection by improving the accuracy and robustness of models while reducing computational costs and memory usage. This makes it highly advantageous for real-world applications with limited resources. Our comparative analysis of various architectures reveals that knowledge distillation consistently achieves superior detection performance, especially in scenarios involving smaller datasets or resource-constrained environments. The ability of distilled models to maintain high detection accuracy while being lightweight and efficient underscores the transformative potential of this approach in advancing object detection technologies. Looking ahead, the future potential of research in this domain is vast and promising. Continued research and innovation in knowledge distillation holds promise of further advancing the capabilities of object detection systems, paving the way for more intelligent and resource-efficient solutions in various applications.

## Figures and Tables

**Figure 1 sensors-26-00292-f001:**
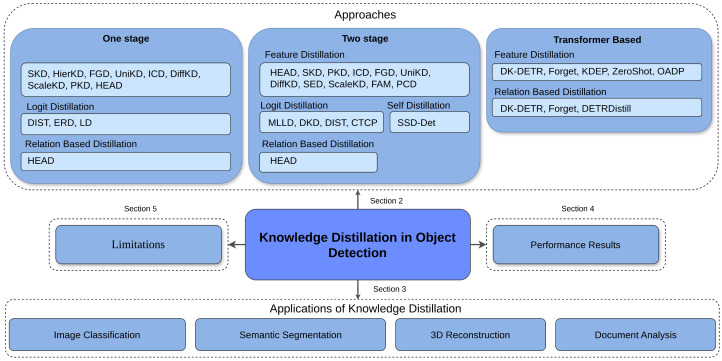
Overview of knowledge distillation survey: This figure summarizes the different sections of the survey along with their references. It also briefly mentions the approaches discussed in this paper, categorized according to the type of object detector on which knowledge distillation has been applied. Specifically, Section 2 reviews the various approaches, Section 3 discusses applications such as Image Classification, Segmentation, 3D Reconstruction, and Document Analysis, Section 4 presents performance results, and Section 5 highlights the limitations of current methods.

**Figure 2 sensors-26-00292-f002:**
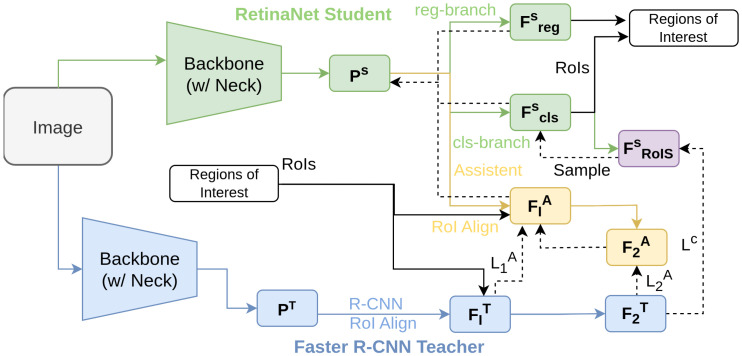
Overview of HEAD [120]: Faster R-CNN is the teacher and RetinaNet the student. An assistant resembling the teacher’s head receives student backbone features alongside the student head. The teacher head processes its own features.

**Figure 3 sensors-26-00292-f003:**
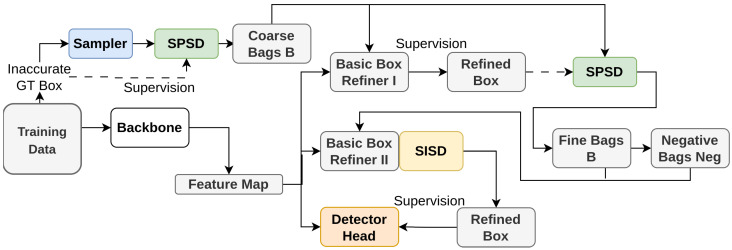
Framework of SSD-Det [126]: The framework includes a box refiner, SPSD, SISD, and a detector head. SPSD refines proposals via MIL training, guided by neighborhood sampling around noisy annotations. SISD predicts IoU and combines it with classification scores for better box selection. SPSD shares the detector backbone.

**Figure 4 sensors-26-00292-f004:**
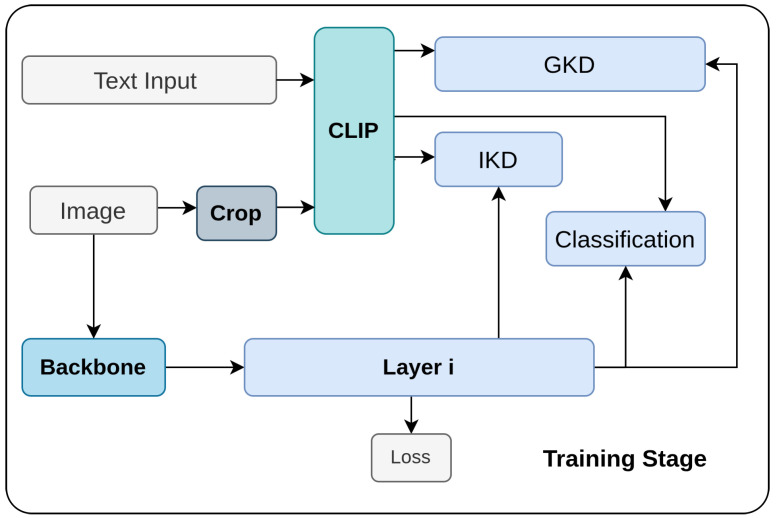
Overview of HierKD [135]: An open-vocabulary one-stage detector applies hierarchical visual-language distillation. IKD aligns features with CLIP visuals, GKD aligns multi-layer features with captions, and CLIP embeddings are replaced with novel categories during inference.

**Figure 5 sensors-26-00292-f005:**
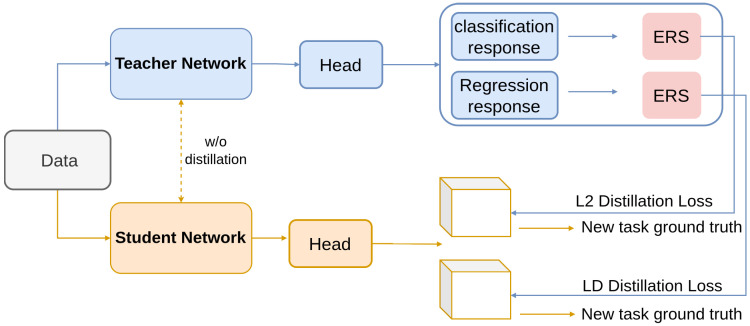
Framework of ERD [117]: Overall architecture of elastic response distillation for incremental object detection. Different colors are used to differentiate between the student and teacher networks.

**Figure 6 sensors-26-00292-f006:**
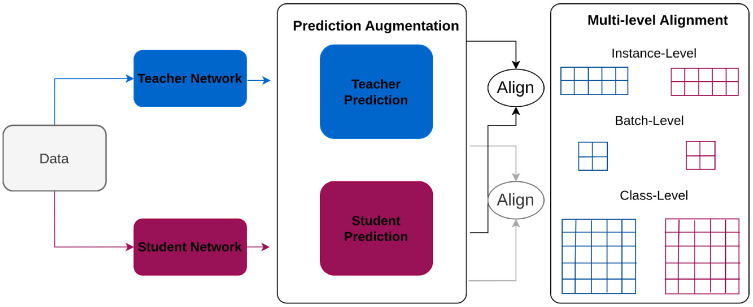
Overview of MLLD [116]: Teacher and student predictions are augmented with multiple temperatures and aligned at instance, batch, and class levels for comprehensive knowledge transfer.

**Figure 7 sensors-26-00292-f007:**
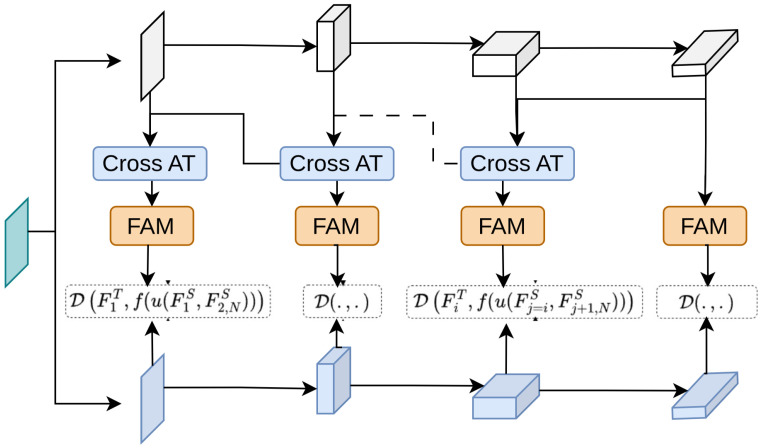
Overview of FAM [150]: CrossAT denotes cross-attention, FAM is the proposed frequency attention module, and D is the distance function. $F^{T}$ and $F^{S}$ are the teacher’s and student’s feature maps. The triangles and pentagrams denote feature representations at different hierarchical stages of the teacher and student networks, respectively.

**Figure 8 sensors-26-00292-f008:**
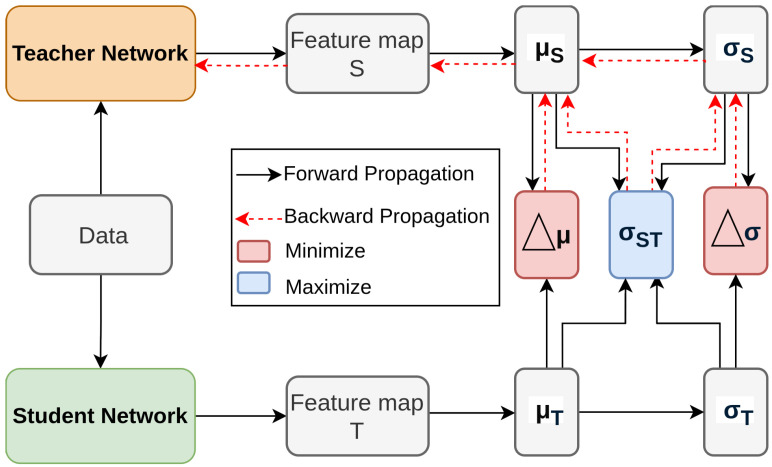
Overview of SKD [113]: Distill relational knowledge using local mean (μ), variance ($\sigma^{2}$), and cross-correlation ($\sigma^{ST}$) between feature spaces.

**Figure 9 sensors-26-00292-f009:**
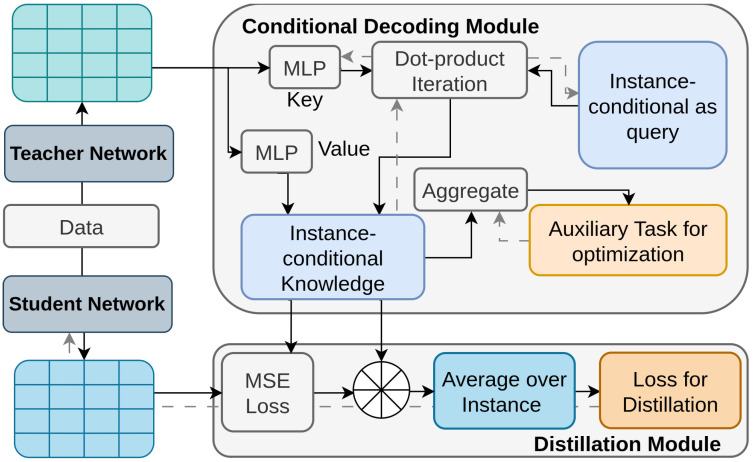
Overview of ICD [118]: A decoding module is proposed to retrieve knowledge via query-based attention, with instance annotations as queries. An auxiliary task optimizes the decoder, and attention-weighted feature distillation updates the student. Gray arrows represent backward propagation, and black arrows represent forward propagation.

**Figure 10 sensors-26-00292-f010:**
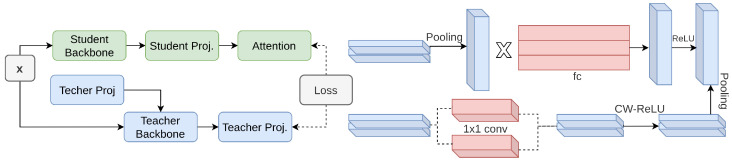
Architecture of PCD [155]: The teacher’s projection head is adapted with SpatialAdaptor. Distillation loss is the average contrastive loss over corresponding student–teacher pixel pairs. Right panel shows the teacher head workflow before (**top**) and after (**bottom**) applying SpatialAdaptor, with the rightmost pooling layer illustrating its invariance.

**Figure 11 sensors-26-00292-f011:**
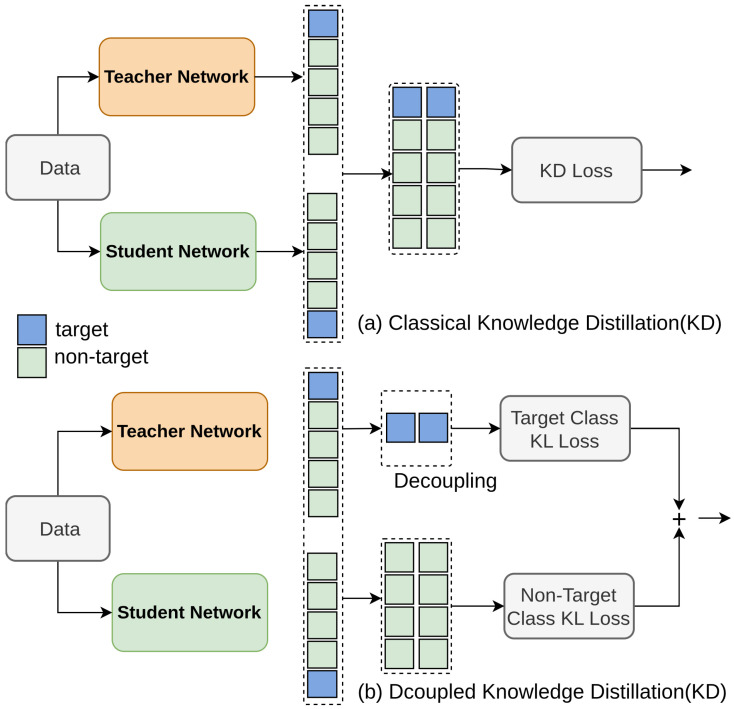
Illustration of classical KD and proposed DKD [115]: KD is reformulated as a weighted sum of TCKD and NCKD, where coupling with the teacher’s target confidence and the part weights limits effectiveness and flexibility. DKD addresses this using hyperparameters α (TCKD) and β (NCKD) to improve knowledge transfer.

**Figure 12 sensors-26-00292-f012:**
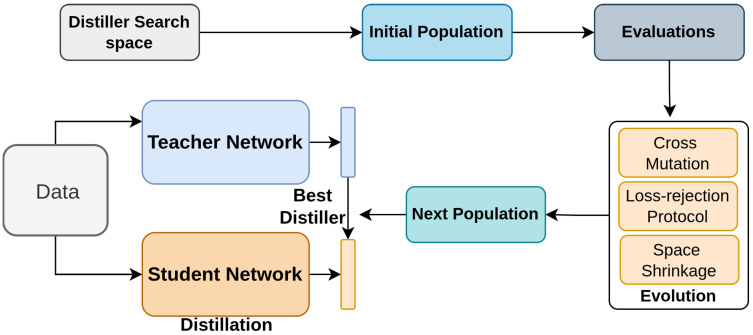
Overview of KD-Zero [119]: Candidate distillers are randomly sampled and evaluated on validation performance and the teacher–student gap. Weak distillers are removed, and cross mutation generates new candidates. The best-performing distiller is then selected for distillation.

**Figure 13 sensors-26-00292-f013:**
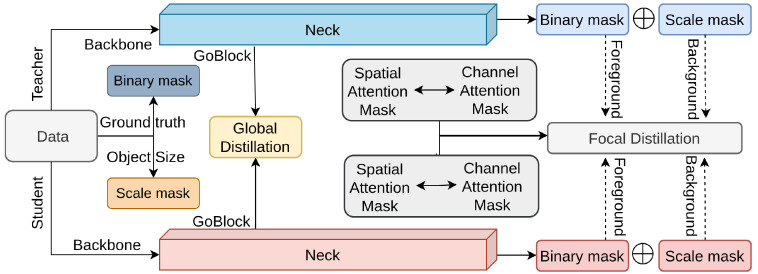
Overview of FGD [124]: Focal distillation separates foreground and background, guiding the student to focus on key teacher features. Global distillation aligns the overall context between student and teacher.

**Figure 14 sensors-26-00292-f014:**
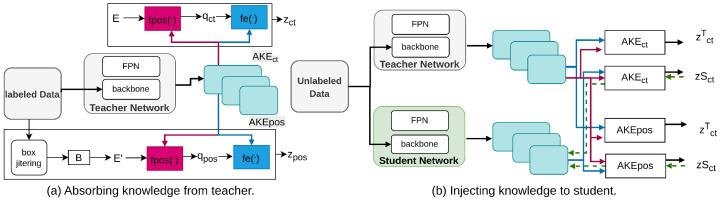
Overview of UniKD [163]: (**a**) Pretrain AKE modules with the teacher fixed. (**b**) During distillation, AKE modules are fixed, and only the student is updated using teacher-based queries.

**Figure 15 sensors-26-00292-f015:**
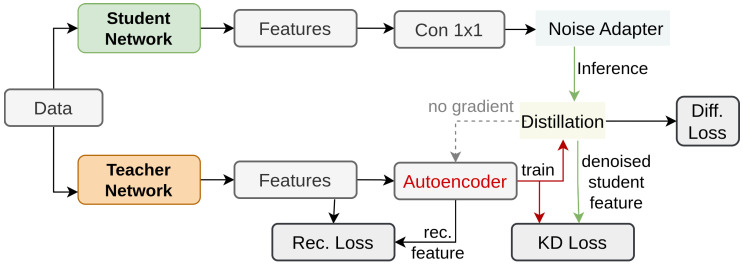
Overview of DiffKD [166]: With Bottleneck referring to the ResNet Bottleneck block.

**Figure 16 sensors-26-00292-f016:**
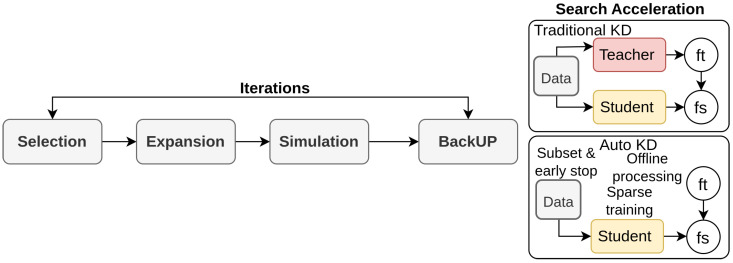
Framework of Auto-KD [14]: The search space is modeled as an MCT, with Monte Carlo Tree Search identifying the optimal distillation design and search acceleration strategies applied.

**Figure 17 sensors-26-00292-f017:**
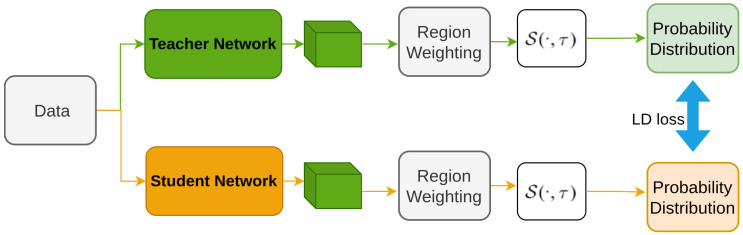
Illustration of LD [172]: Only the localization branch is shown. Bounding boxes are converted to probability distributions, weighted by key regions, and the LD loss is computed between teacher and student.

**Figure 18 sensors-26-00292-f018:**
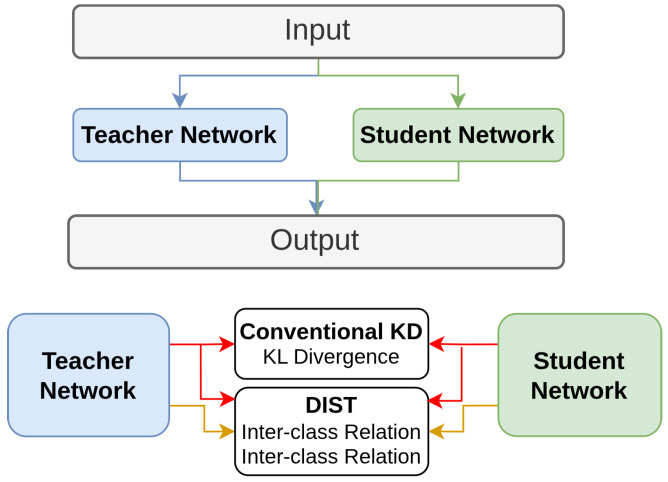
Overview of proposed DIST vs existing KD methods [160]: Conventional KD aligns student and teacher outputs point-wise, while instance relation methods match feature-level correlations. DIST preserves inter-class correlations and intra-class correlations.

**Figure 19 sensors-26-00292-f019:**
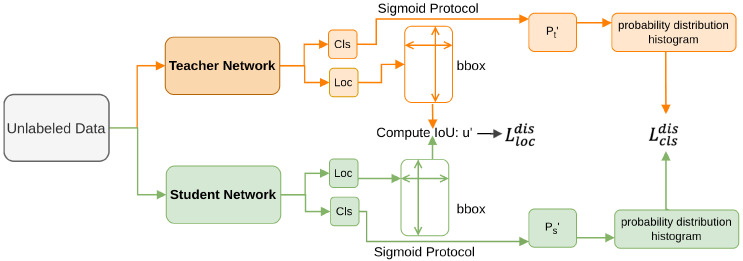
Pipeline of CTCP method [176]: Binary Classification Distillation Loss converts classification logits into multiple binary maps and applies a binary cross-entropy-style loss; IoU-based Localization Distillation Loss transfers localization knowledge by computing IoUs between teacher and student predicted boxes.

**Figure 20 sensors-26-00292-f020:**
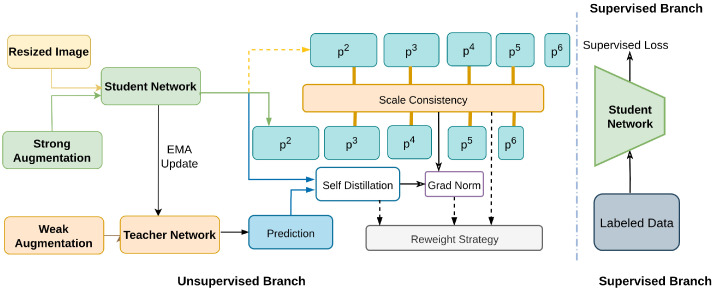
Overview of SED [177]: Conventional KD aligns student ($s\in R^{5}$) and teacher ($t\in R^{5}$) outputs point-wise, while instance relation methods match feature-level correlations.

**Figure 21 sensors-26-00292-f021:**
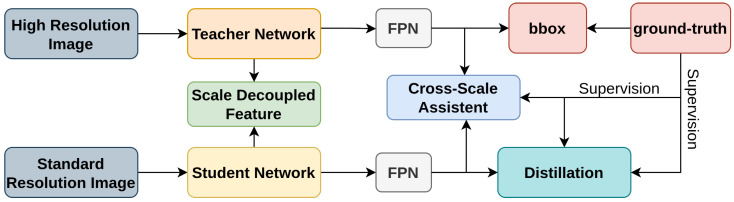
The overview of ScaleKD [182]: It includes a scale-decoupled feature distillation module and a cross-scale assistant module to enhance small object detection.

**Figure 22 sensors-26-00292-f022:**
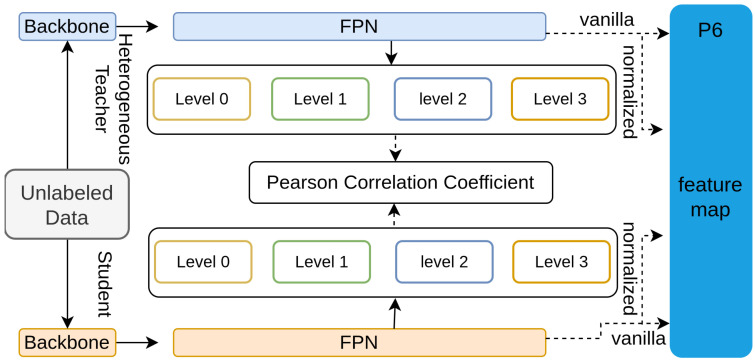
Overview of the PKD [114]: Using Pearson Correlation, FPN features are normalized to align student and teacher activation patterns, improving distillation effectiveness.

**Figure 23 sensors-26-00292-f023:**
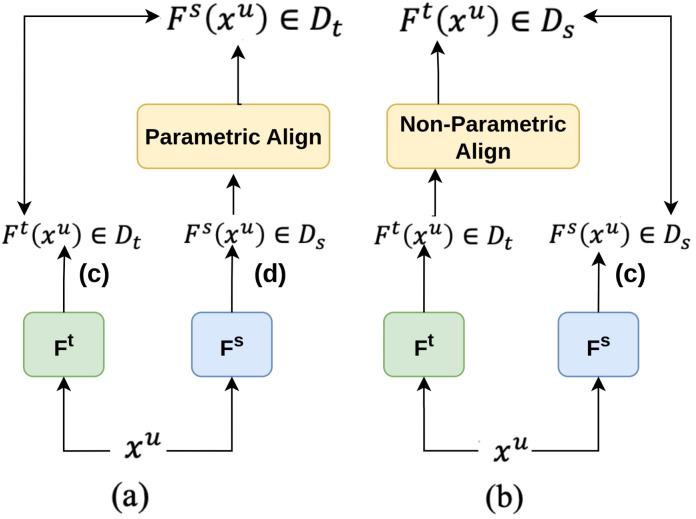
Overview of KDEP framework with (**a**) parametric aligning and (**b**) non-parametric aligning [185].

**Figure 24 sensors-26-00292-f024:**
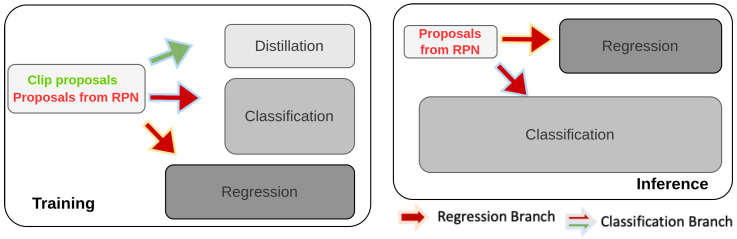
Overview of ZeroShot [193]: The RoI head of the two-stage detector uses per-CLIP proposal distillation weights and a semantic-based regressor to enhance performance.

**Figure 25 sensors-26-00292-f025:**
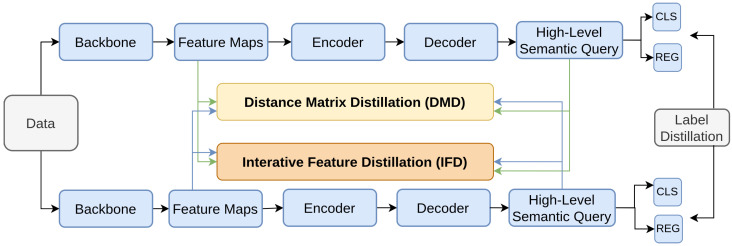
Overview of knowledge distillation within-class and between-class [194].

**Figure 26 sensors-26-00292-f026:**
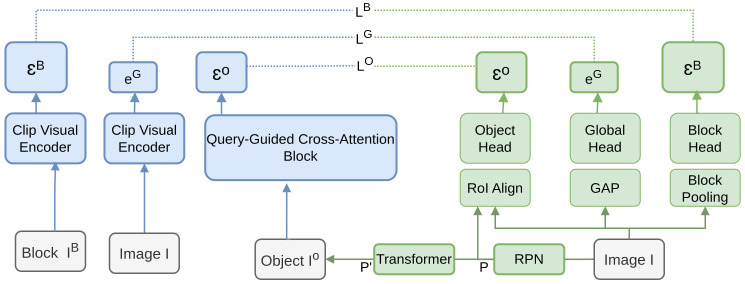
Training pipeline for OADP [197]: OADP uses a pyramid with global, block, and object distillation. RPN proposals generate object embeddings via RoI Align and masked attention with a token. Global and block embeddings are extracted via pooling, with teacher embeddings from CLIP.

**Figure 27 sensors-26-00292-f027:**
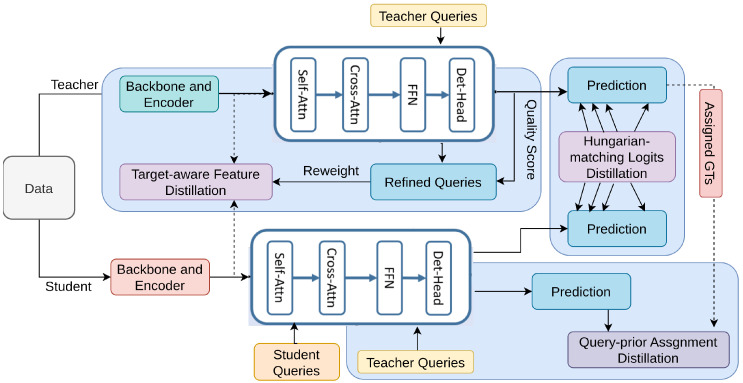
Overview of DETRDistill [198]: The approach includes a transformer-based teacher with a large backbone, a lightweight student detector, and three distillation modules: (i) Hungarian-matching logits distillation, (ii) target-aware feature distillation, and (iii) query-prior assignment distillation, omitting original supervision for clarity.

**Figure 28 sensors-26-00292-f028:**
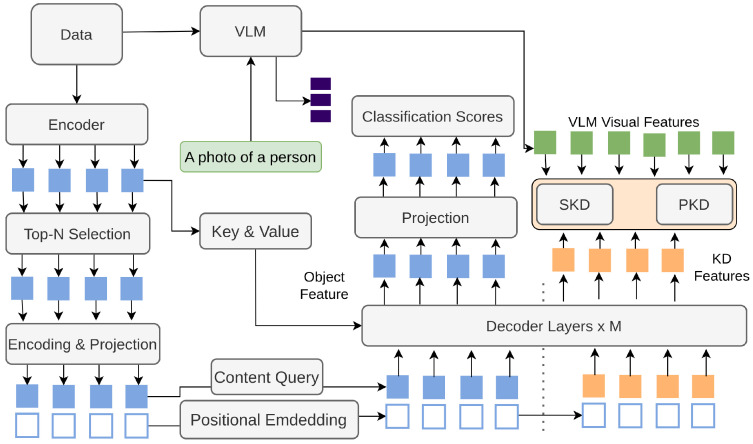
Overview of DK-DETR [202]: DK-DETR extracts object features via a Deformable DETR encoder–decoder and matches them with VLM text embeddings for detection. A training-only distillation branch transfers knowledge from the VLM image encoder to aid novel categories.

**Figure 29 sensors-26-00292-f029:**
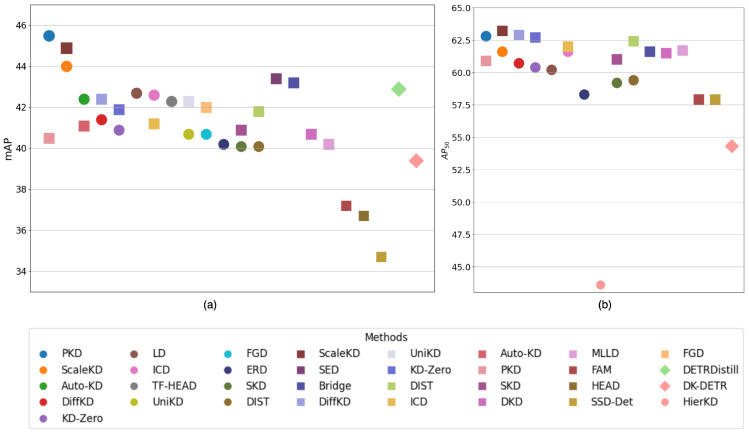
Comparisons of mAP and $AP_{50}$ of all the discussed strategies. The figure visualizes the performance of different knowledge distillation techniques applied in object detection. (**a**) provides a comparison of the performance between different strategies through mAP whereas (**b**) provides a similar comparison using $AP_{50}$. In each graph, the circles represent One-Stage, squares Two-Stage, and diamonds End-to-End strategies, respectively, thus encompassing all the categories of object detectors where knowledge distillation has been employed.

**Table 1 sensors-26-00292-t001:** Object detection performance on MS COCO dataset. Comparison of object detection methods across different stages on the MS COCO dataset.

Methods	Stages	Reference	Teacher	Student	COCO		
					*mAP*	*AP* _50_	*AP* _75_
HEAD [120]	One-Stage	ECCV22	Faster R50	RetinaNet R18 (31.7)	36.2	-	-
DIST [160]		NeurIPS22	RetinaNet-X101	RetinaNet-R50 (37.4)	40.1	59.4	23.2
SKD [113]		NeurIPS22	RetinaNet-R101	RetinaNet-R50 (36.4)	40.1	59.2	43.1
HierKD [135]		CVPR22	CLIP-R50	ATSS	-	43.6	-
ERD [117]		CVPR22	GFLV1 R-50	GFLV1 R-50	40.2	58.3	43.6
FGD [124]		CVPR22	RetinaNet-X101	RetinaNet-R50 (37.4)	40.7	-	-
UniKD [163]		ICCV23	RetinaNet-X101	RetinaNet-R50 (37.4)	40.7	-	-
ICD [118]		NeurIPS21	FCOS R101	FCOS R50	42.6	61.6	45.8
LD [172]		CVPR22	RetinaNet-R101	RetinaNet-R50 (36.9)	42.7	60.2	46.7
KD-Zero [119]		NeurIPS23	RetinaNet-X101	RetinaNet-R50 (37.4)	40.9	60.4	43.5
DiffKD [166]		NeurIPS23	RetinaNet-X101	RetinaNet-R50 (37.4)	41.4	60.7	44.0
Auto-KD [14]		ICCV23	RetinaNet-X101	RetinaNet-R50 (37.4)	41.1	-	-
ScaleKD [182]		CVPR23	FOCS R101	FOCS R50 (38.5)	44.0	61.6	41.6
PKD [114]		NeurIPS22	TOOD-ResX101	TOOD-Res50 (42.4)	45.5	62.8	49.3
HEAD [120]	Two-Stage	ECCV22	Cascade R50	Faster R18 (33.9)	36.7	-	-
FAM [150]		WACV24	Faster R101	Faster R50 (37.9)	40.8	61.4	44.5
MLLD [116]		CVPR23	Faster R101	Faster R50 (37.9)	40.2	61.7	44.6
DKD [115]		CVPR22	Faster R101	Faster R50 (37.9)	40.7	61.5	44.4
SKD [113]		NeurIPS22	Faster ResNeXt101	Faster R50 (37.4)	40.9	61.0	44.9
PKD [114]		NeurIPS22	Faster R101	Faster R50 (37.9)	40.5	60.9	44.4
Auto-KD [14]		ICCV23	Cascade R101	Faster R50 (38.4)	42.4	-	-
ICD [118]		NeurIPS21	Faster R101	Faster R50 (37.9)	40.9	-	-
DIST [160]		NeurIPS22	Cascade R101	Faster R50 (38.4)	41.8	62.4	45.6
KD-Zero [119]		NeurIPS23	Cascade R101	Faster R50 (38.4)	41.9	62.7	45.5
FGD [124]		CVPR22	Cascade-X101	Faster-R50 (38.4)	42.0	-	-
UniKD [163]		ICCV23	Cascade R101	Faster R50 (38.4)	42.3	-	-
DiffKD [166]		NeurIPS23	Faster R101	Faster R50 (38.4)	42.4	62.9	46.4
CTCP [176]		ICCV23	GFocal R101	GFocal R50 (40.1)	43.2	61.6	46.9
SED [177]		CVPR22	Faster R50	Faster R50 (38.40)	43.4	-	-
ScaleKD [182]		CVPR23	Cascade R101	Cascade R50 (41.0)	44.9	63.2	48.8
DK-DETR [202]	Transformer-Based	ICCV23	VLM	Deformable DETR	39.4	54.3	43.0
Forget [194]		ICCV23	Deformable DETR	Deformable DETR	39.8	-	-
DETRDistill [198]		ICCV23	AdaMixer ResNet-101	AdaMixer ResNet-50 (42.3)	44.7	-	-

**Table 2 sensors-26-00292-t002:** Advantages and limitations.

Methods	Advantages	Limitations
HEAD [120]	Effectively bridges the semantic gap between heterogeneous teacher–student detectors	Requires customized algorithm designs for each teacher–student pair, limiting its flexibility
DK-DETR [202]	Significantly improves open-vocabulary object detection performance for novel categories without degrading the performance of base categories	Depends on auxiliary queries used solely during training introducing additional complexity and abstraction
FAM [150]	Captures global information, enhancing the student’s ability to mimic the teacher’s features	Needs additional computational complexity due to the use of FFT and inverse FFT operations
MLLD [116]	Introduces multi-level prediction alignment to enhance logit distillation	Computational complexities and additional hyperparameters present, affecting its practicality for large-scale tasks and industrial applications
DKD [115]	Improves the effectiveness of logit distillation by decoupling the distillation of target and non-target class knowledge	The use of this approach is complicated in industrial applications due to the increase in the number of hyperparameters
DIST [160]	Enhances knowledge distillation by leveraging both inter-class and intra-class relational knowledge through a correlation-based loss	Mainly targeted homogeneous teacher–student model pairs, leaving cross-architecture distillation largely unexplored
SKD [113]	Uses $l_{SSIM}$ to capture additional relational knowledge in the feature space taking into account the spatial relationships	Its effectiveness may be influenced by specific experimental setups
HierKD [135]	Enables the detection of both seen and unseen categories, improving performance in open-vocabulary settings	Requires substantial computational resources due to its dependence on large-scale pre-trained vision-language models
ERD [117]	Highly effective in preserving knowledge from old classes during incremental object detection	Limited by its heavy reliance on low-level feature selection and the underutilization of high-level semantic information
Forget [194]	Excels in retaining knowledge from old classes during incremental object detection	Adapting this method to CNN-based detectors requires careful design to balance performance and computational efficiency
FGD [124]	Effectively focuses on critical pixels and channels while incorporating global contextual information	Relies on precise ground-truth bounding boxes to separate foreground and background
UniKD [163]	Transfers knowledge between diverse teacher–student pairs without complex adjustments	Learning from advanced teachers with different architectures does not always improve performance
ICD [118]	Able to effectively transfer knowledge useful for both classification and localization of every instance	This approach has high complexity and computational cost, especially during training
LD [172]	Improves dense object detection by transferring fine-grained localization knowledge	Relies on a high-performing teacher model, which may not always be available or feasible to train.
KD-Zero [119]	Automatically discovers optimal distillation strategies for any teacher–student pairs	Dependent on expert knowledge for initial setup and struggles with mismatched architectural styles between teacher and student models.
DiffKD [166]	Aligns student and teacher features through denoising, improving performance across tasks and models.	High computational cost due to the denoising process diffusion models
CTCP [176]	Improves performance by addressing protocol inconsistencies without additional costs	The approach reduces classification errors but does not contribute to reducing localization errors
SED [177]	Effectively handles large object size variance and class imbalance in semi-supervised object detection.	Increased complexity and computational demands impact scalability and practical implementation
DETRDistill [198]	Enhances performance across different DETR models	Requires additional training complexity required to adapt the distillation framework to various DETR models
Auto-KD [14]	Efficient selection of distillation strategies using Monte Carlo Tree Search	Encounter difficulties in adapting these strategies to a wide range of model architectures and tasks
ScaleKD [182]	Enhances small object detection by transferring scale-aware knowledge into the student model	Struggles to handle scale-aware knowledge transfer for objects of sizes not well-represented in the training data
PKD [114]	focuses on relational information between features, reducing the impact of magnitude differences	Corruption of features from downsampling in the teacher model, hindering the student model’s ability to imitate specific information

## Data Availability

No new data were created in this study. Data sharing is not applicable to this article.

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
