# Peer review of "Knowledge Distillation in Object Detection: A Survey from CNN to Transformer"

_sensors, 2026, doi:10.3390/s26010292_

Round 1

Reviewer 1 Report

Comments and Suggestions for Authors

This research presents a comprehensive survey on the application of Knowledge Distillation (KD) for compressing object detection models, tracking the progression from convolutional neural networks (CNNs) to modern transformer-based architectures.  

Herein the comments for this manuscript:

  1. What is the main purpose of Knowledge Distillation?
  2. What are the three categories of object detectors?
  3.  It ius suggested to include this research in the introduction:

Elhanashi, A.; Dini, P.; Saponara, S.; Zheng, Q. Integration of Deep Learning into the IoT: A Survey of Techniques and Challenges for Real-World Applications. Electronics 202312, 4925. https://doi.org/10.3390/electronics12244925

  1. What problem does the HEAD framework solve?
  2. What imbalance does the FGD method address?
  3. Name one other computer vision task that uses Knowledge Distillation.
  4. What problem does the ERD method tackle?
  5. Does DKD couple or decouple target and non-target class knowledge?
  6. Which method uses evolutionary algorithms for automation?
  7. Why is Knowledge Distillation more challenging for object detection than for image classification?
  8. What is the future direct?

Author Response

Here the comments for this manuscript:

  1. What is the main purpose of Knowledge Distillation?

Knowledge Distillation (KD)is a pivotal technique in deep learning for transferring knowledge from large, high-capacity teacher models to smaller, more efficient student models. Instead of relying solely on hard labels, the student leverages the teacher’s outputs, including soft probability distributions and intermediate representations, as guidance. The objective is to produce a model that is computationally lightweight while retaining strong performance, making KD especially valuable for deployment on resource-constrained devices.

  1. What are the three categories of object detectors?

Object detection methods are divided into three categories:

  1. Two-Stage Detectors
    1. First generate region proposals (candidate object locations), then classify and refine them.
    2. Example: R-CNN, Fast R-CNN, Faster R-CNN, Mask R-CNN.
  2. One-Stage Detectors
    1. Skip the region proposal step and directly predict bounding boxes and class probabilities over dense locations.
    2. Example: YOLO series, SSD, RetinaNet.
  3. Transformer-based Detectors
    1. Use transformers for detection without predefined anchors or proposals.
    2. Example: DETR (Detection Transformer), Deformable DETR, DINO-DETR.
  4.  It is suggested to include this research in the introduction:

Elhanashi, A.; Dini, P.; Saponara, S.; Zheng, Q. Integration of Deep Learning into the IoT: A Survey of Techniques and Challenges for Real-World Applications. Electronics 2023, 12, 4925. https://doi.org/10.3390/electronics12244925

 We have added the paper.

  1. What problem does the HEAD framework solve?
    The HEAD framework effectively bridges the semantic gap between heterogeneous detectors, enabling efficient knowledge transfer and improving the performance of lightweight object detectors.
  2. What imbalance does the FGD method address?
    The imbalance between foreground and background features in object detection can hinder effective knowledge distillation.
  1. Name one other computer vision task that uses Knowledge Distillation.

Object detection, pose estimation, super-resolution, video understanding, depth estimation, salient object detection, medical imaging, optical flow estimation, and generative tasks

  1. What problem does the ERD method tackle?

The Elastic Response Distillation (ERD) method tackles catastrophic forgetting in incremental object detection, ensuring that a student model retains knowledge of previously learned classes while learning new ones.

  1. Does DKD couple or decouple target and non-target class knowledge?

Unlike traditional KD methods that couple these two components, DKD separates them into TCKD (target class knowledge distillation) and NCKD (non-target class knowledge distillation), allowing each to be optimized independently for better training efficiency and performance.

  1. Which method uses evolutionary algorithms for automation?

The method that uses evolutionary algorithms for automation is KD-Zero. It is designed to automate the search for optimal knowledge distillation strategies by decomposing distillation into components (knowledge transformations, distance functions, and loss weights) and applying evolutionary search to find the best combination

  1. Why is Knowledge Distillation more challenging for object detection than for image classification?

KD for object detection is more challenging because it involves multiple predictions per image, dual tasks classification and localization, imbalanced foreground-background regions, heterogeneous architectures, and spatial-context dependencies, whereas image classification deals with a single label per image and a simpler output space.

  1. What is the future direct?

The future direction in this field focuses on further improving object detection systems through advanced knowledge distillation techniques. Ongoing research aims to make these systems more intelligent, accurate, and resource-efficient

Reviewer 2 Report

Comments and Suggestions for Authors

Comprehensive Evaluation:  
This manuscript offers an extensive survey on the application of knowledge distillation (KD) within the domain of object detection, tracing the progression from convolutional neural network (CNN)-based techniques to Transformer-based methodologies. Additionally, it briefly addresses KD applications in other fields, including image classification, semantic segmentation, 3D reconstruction, and document analysis. The topic is both timely and pertinent, given the growing importance of KD for deploying deep learning models in environments with limited computational resources. The manuscript is well-organized and substantiated with numerous recent references, effectively reflecting the current state of research. Nonetheless, several significant issues warrant revision. The following detailed recommendations are proposed:  

1. The title of the manuscript does not accurately represent its content. Approximately seventeen pages are dedicated to reviewing advances in knowledge distillation for object detection, whereas other domains such as classification, semantic segmentation, 3D reconstruction, and document analysis receive only about two pages of coverage. It is advisable either to revise the title to more precisely reflect the primary focus—for example, “Knowledge Distillation in Object Detection: …”—or to expand the discussion of these additional domains to ensure a more balanced treatment.  

2. The manuscript exhibits deficiencies in logical coherence. The review predominantly enumerates methods without sufficient transitional elements between sections. It is recommended to incorporate summary paragraphs at the conclusion of each methodological category to elucidate their respective strengths and limitations. Furthermore, the transition from “CNN-Based” to “Transformer-Based” approaches lacks an explanation of the underlying drivers of this technological shift, such as the influence of self-attention mechanisms on knowledge representation. Including a comparative analysis of their representational capacities would enhance the narrative linking technological developments.  

3. The manuscript does not provide an in-depth comparative analysis of the surveyed methods. For various KD architectures in object detection, the evaluation is limited to listing average precision (AP) metrics. It is suggested to augment this with an analysis of the commonalities and distinctions among different distillation strategies, considering aspects such as performance, applicable scenarios, and computational complexity.  

4. The Limitations section is insufficiently detailed regarding the different categories of methods. Providing a comparative summary of the advantages and disadvantages of the reviewed approaches—categorized as one-stage, two-stage, and Transformer-based methods—would offer clearer guidance to researchers in selecting appropriate techniques.  

5. The manuscript contains several grammatical inaccuracies, including the use of “limitations” on line 113 and “Its effectiveness” in Table 2. A thorough language revision is recommended to improve clarity and readability.

Author Response

Comprehensive Evaluation: 
This manuscript offers an extensive survey on the application of knowledge distillation (KD) within the domain of object detection, tracing the progression from convolutional neural network (CNN)-based techniques to Transformer-based methodologies. Additionally, it briefly addresses KD applications in other fields, including image classification, semantic segmentation, 3D reconstruction, and document analysis. The topic is both timely and pertinent, given the growing importance of KD for deploying deep learning models in environments with limited computational resources. The manuscript is well-organized and substantiated with numerous recent references, effectively reflecting the current state of research. Nonetheless, several significant issues warrant revision. The following detailed recommendations are proposed:  

  1. The title of the manuscript does not accurately represent its content. Approximately seventeen pages are dedicated to reviewing advances in knowledge distillation for object detection, whereas other domains such as classification, semantic segmentation, 3D reconstruction, and document analysis receive only about two pages of coverage. It is advisable either to revise the title to more precisely reflect the primary focus—for example, “Knowledge Distillation in Object Detection: …”—or to expand the discussion of these additional domains to ensure a more balanced treatment.  

Change the name as Knowledge Distillation in Object Detection: A Survey from CNN to Transformer

  1. The manuscript exhibits deficiencies in logical coherence. The review predominantly enumerates methods without sufficient transitional elements between sections. It is recommended to incorporate summary paragraphs at the conclusion of each methodological category to elucidate their respective strengths and limitations. Furthermore, the transition from “CNN-Based” to “Transformer-Based” approaches lacks an explanation of the underlying drivers of this technological shift, such as the influence of self-attention mechanisms on knowledge representation. Including a comparative analysis of their representational capacities would enhance the narrative linking technological developments. 

Thank you for the comment. We added category summaries and clarified the shift from CNN- to Transformer-based methods with a comparative discussion of their representational capacities as subsection 2.1, 2.2.

  1. The manuscript does not provide an in-depth comparative analysis of the surveyed methods. For various KD architectures in object detection, the evaluation is limited to listing average precision (AP) metrics. It is suggested to augment this with an analysis of the commonalities and distinctions among different distillation strategies, considering aspects such as performance, applicable scenarios, and computational complexity.  

Thank you for pointing this out. We have revised section 4 to include the analysis of different distillation strategies.

  1. The Limitations section is insufficiently detailed regarding the different categories of methods. Providing a comparative summary of the advantages and disadvantages of the reviewed approaches—categorized as one-stage, two-stage, and Transformer-based methods—would offer clearer guidance to researchers in selecting appropriate techniques.  

Thank you for pointing this out. We have revised the Limitations section to include a more detailed comparative summary of one-stage, two-stage, and Transformer-based methods.

  1. The manuscript contains several grammatical inaccuracies, including the use of “limitations” on line 113 and “Its effectiveness” in Table 2. A thorough language revision is recommended to improve clarity and readability.

Thank you for pointing this out. The grammatical inaccuracies have been corrected, including the use of “limitations” on line 113 and “Its effectiveness” in Table 2.

Reviewer 3 Report

Comments and Suggestions for Authors

The survey paper provides a comprehensive understanding of the current state of research in this domain and highlights the significant advancements made possible through knowledge distillation techniques. However, the methodology of the survey should be reproducible. The authors should employ a systematic review approach to reorganize their findings.

Author Response

The survey paper provides a comprehensive understanding of the current state of research in this domain and highlights the significant advancements made possible through knowledge distillation techniques. However, the methodology of the survey should be reproducible. The authors should employ a systematic review approach to reorganize their findings.

Thank you for pointing this out. We have added the explanation about this in the last paragraph of the introduction.

Round 2

Reviewer 1 Report

Comments and Suggestions for Authors

The paper can be accepted for publication.

Author Response

We thank the reviewer for the positive assessment and recommendation for acceptance. Minor edits were made to further improve clarity and presentation.

Reviewer 3 Report

Comments and Suggestions for Authors

There are references that are not cited correctly as "222." The authors need to review the entire document to ensure accuracy.

Author Response

We thank the reviewer for the positive evaluation of our work. We appreciate the comment regarding incorrect references. In the revised manuscript, we thoroughly reviewed the entire reference list and corrected all issues, including the previously noted citation “222.” We also ensured consistency and accuracy across all citations.